# One-Shot Pruning of Recurrent Neural Networks by Jacobian Spectrum Evaluation

**Matthew Shunshi Zhang**
University of Toronto
`matthew.zhang@mail.utoronto.ca`

**Bradly C. Stadie**
Vector Institute

## Abstract

Recent advances in the sparse neural network literature have made it possible to prune many large feed forward and convolutional networks with only a small quantity of data. Yet, these same techniques often falter when applied to the problem of recovering sparse recurrent networks. These failures are quantitative: when pruned with recent techniques, RNNs typically obtain worse performance than they do under a simple random pruning scheme. The failures are also qualitative: the distribution of active weights in a pruned LSTM or GRU network tend to be concentrated in specific neurons and gates, and not well dispersed across the entire architecture. We seek to rectify both the quantitative and qualitative issues with recurrent network pruning by introducing a new recurrent pruning objective derived from the spectrum of the recurrent Jacobian. Our objective is data efficient (requiring only 64 data points to prune the network), easy to implement, and produces 95 % sparse GRUs that significantly improve on existing baselines. We evaluate on sequential MNIST, Billion Words, and Wikitext.

## 1 Introduction

Within the neural network community, network pruning has been something of an evergreen problem. There are several motivations for pruning a neural network. Theoretically, overparameterization is a well known but poorly understood quality of many networks. Pruning algorithms provide a link between overparameterized models appropriately parameterized models. Thus, these algorithms may provide insights into exactly why overparameterized models have so much success. Indeed, recent work has closely linked the efficient utilization of model capacity with generalization results (Arora et al., 2018). From a more practical perspective, overparameterized networks require more storage capacity and are computationally more expensive than their pruned counterparts. Hence, there is an incentive to use pruned networks rather than fully dense networks during deployment.

For years, many of the most successful network pruning techniques were iterative — relying on cycle of pruning and retraining weights to induce sparsity in the network. As identified in Lee et al. (2018), these methods usually either enforce a sparsity-based penalty on the weights (Han et al., 2015; LeCun et al., 1990), or else prune based on some fitness criteria (Carreira-Perpinán & Idelbayev, 2018; Chauvin, 1989). Recent advances in pruning literature suggest that such costly cycles of pruning and retraining might not always be necessary. For some problems, there exists a small subnetwork within the original larger network such that training against this smaller network produces comparable performance to training against the original fully dense network. The Lottery Ticket Hypothesis (Frankle & Carbin, 2019) provides a method for recovering these networks, but only after training is complete. SNIP (Lee et al., 2018) and GraSP (Wang et al., 2020) provide a saliency criterion for identifying this small subnetwork using less than 100 data points, no training, and no iterative pruning.

Our present work began by asking the question: "How well do these newly discovered pruning techniques, which optimize a network sensitivity objective, work on recurrent neural networks?" Although Lee et al. (2018) does evaluate the SNIP pruning criterion on both GRU and LSTM networks, we found these results to be somewhat incomplete. They did not provide a comparison to random pruning, and the chosen tasks were not extensive enough to draw definitive conclusions. When compared against random pruning, we found that the SNIP and GraSP pruning objective

performed similarly to or worse than random pruning. This left us wondering where those techniques were falling short, and if a better pruning objective could be developed that takes the temporal structure of recurrent networks into account.

In this paper, we propose a new pruning objective for recurrent neural networks. This objective is based on recent advances in mean field theory (Gilboa et al., 2019; Chen et al., 2018a), and can be interpreted as forcing the network to preserve weights that propagate information through its temporal depths. Practically, this constraint is imposed by forcing the singular values of the temporal Jacobian with respect to the network weights to be non-degenerate. We provide a discussion about the similarities and differences between our objective and the SNIP and GraSP pruning objectives. It can be shown that these prior objectives fail to ensure that the temporal Jacobian of the recurrent weights is well conditioned. Our method is evaluated with a GRU network on sequential MNIST, Wikitext, and Billion Words. At 95% sparsity, our network achieves better results than fully dense networks, randomly pruned networks, SNIP (Lee et al., 2018) pruned networks, and GraSP (Wang et al., 2020) pruned networks.

## 2 PRUNING RECURRENT NETWORKS BY JACOBIAN SPECTRUM EVALUATION

### 2.1 NOTATION

We denote matrices and vectors by upper- and lower-case bold letters respectively. Vector-valued functions are bolded, whereas scalar valued functions are not. Distributions over variables are with the following script: $\mathcal{D}, \mathcal{P}$. We denote the standard $\ell_p$ norm of a vector by $\|\cdot\|_p$. Let $[\cdot]_{ij}$ be the (i,j)-th element of a matrix, and $[\cdot]_i$ the i-th element of a vector. $\vec{1}, \vec{0}$, denotes a vector of 1s or 0s of appropriate length, and use $\odot$ denotes a Hadamard product. $I_A$ represents the standard indicator function. For vectors, superscripts are always used for sequence lengths while subscripts are reserved for indexing vector elements.

### 2.2 PRELIMINARIES

#### 2.2.1 RECURRENT MODELS

Let $\mathbf{X} = \left\{\mathbf{x}^{(t)}\right\}_{t=1}^{S}$; with each $\mathbf{x}^{(t)} \in \mathbb{R}^D$. Similarly, let $\hat{\mathbf{Y}} = \left\{\hat{\mathbf{y}}^{(t)}\right\}_{t=1}^{S}$, where each $\mathbf{y}^{(t)} \in \mathbb{R}^O$ is an associated set of outputs, such that each tuple $(\mathbf{X}, \mathbf{Y}) \stackrel{i.i.d.}{\sim} \mathcal{D}$.

Let $\mathbf{M}(\mathbf{x}; \boldsymbol{\theta}) : \mathbb{R}^D \mapsto \mathbb{R}^O$ be a generic model, parameterized by $\boldsymbol{\theta} \in \mathbb{R}^N$, that maps $\mathbf{X}$ onto an output sequence. Define a recurrent model as a mapping done through iterative computation, such that each $(\hat{\mathbf{y}}^{(t)}, \mathbf{h}^{(t)}) = \mathbf{M}(\mathbf{x}^{(\mathbf{t})}, \hat{\mathbf{h}}^{(t-1)}; \boldsymbol{\theta})$ depends explicitly only on the current input and some latent state of the model, $h$.

We define a loss over the entire sequence of outputs as the sum of a non-sequential loss function, $\tilde{L}$, over an entire sequence: $L(\mathbf{M}, \mathbf{X}, \mathbf{Y}) = \sum_{i=1}^{S} \tilde{L}(\hat{\mathbf{y}}^{(t)}, \mathbf{y}^{(t)})$.

We define a sparse model as one where the parameters factorize into $\theta = c \odot w$, with $c \in \{0, 1\}^N$ a binary mask and $w \in \mathbb{R}^N$ the free values, typically trained by gradient descent. We define a $K$-sparse condition on a sparse model $\mathbf{M}$ as the restriction $\|c\|_0 = K$ during the entire training trajectory. A model is optimally $K$-sparse if it minimizes the expected loss, $\mathbb{E}_{\mathcal{D}}[L(\mathbf{M}, \mathbf{X}, \mathbf{Y})]$ after training while also being subject to a $K$-sparse condition.

#### 2.2.2 MEMORY HORIZON

We introduce the following terms: $N$ is the size of the network hidden state $\mathbf{h}$, and $\mathbf{J}_t \in \mathbb{R}^{N \times N}$ is the temporal Jacobian, of the hidden state at time $t + 1$ with respect to the previous hidden state, $\frac{\partial \mathbf{h}^{(t+1)}}{\partial \mathbf{h}^{(t)}}$, and $\sigma_i^{(t)}$ the singular values of said matrix.

To arrive at a one-shot pruning criteria for recurrent neural networks, we consider the impact of the temporal Jacobian on both forward- and backward-propagation.

- (Backpropagation) The formula for backpropagation through time (BPTT), from the loss at time $s$ can be given as:

$$\nabla_{\boldsymbol{\theta}}\tilde{L}(\hat{\mathbf{y}}_s, \mathbf{y}_s) = \underbrace{\left[\tilde{\mathbf{G}}_{\mathbf{h}^{(s)};\boldsymbol{\theta}}^T + \tilde{\mathbf{G}}_{\mathbf{h}^{(s-1)};\boldsymbol{\theta}}^T\mathbf{J}_{s-1} + \ldots + \tilde{\mathbf{G}}_{\mathbf{h}^{(1)};\boldsymbol{\theta}}^T\prod_{t=1}^{s-1}\mathbf{J}_t\right]}_{\tilde{\mathbf{G}}_s} \cdot \nabla_{\mathbf{h}^{(s)}}\tilde{L}(\hat{\mathbf{y}}_s, \mathbf{y}_s)$$

(1)

  Where $\tilde{\mathbf{G}}_{\mathbf{h}^{(t)};\boldsymbol{\theta}}$ is the Jacobian of $\mathbf{h}^{(t)}$ considering only the explicit dependence on $\theta$.

- (Forward Propagation)

  A single time-step of the network under small perturbations yields the following:

$$\mathbf{M}(\mathbf{x}^{(t)}; \mathbf{h}^{(t)} + \boldsymbol{\epsilon}) \approx \mathbf{h}^{(t+1)} + \mathbf{J}_t\boldsymbol{\epsilon}$$

(2)

  With additional powers of the Jacobian appearing as we observe the entire sequence.

From Equation 1, it can easily be seen that increasing the normed singular values of each $\mathbf{J_t}$ will on average exponentially increase the gradient signal from later sequence elements, which will expedite convergence by reducing the vanishing gradient problem. From Equation 2, we additionally note that a well-conditioned Jacobian would enable the network to preserve separation of distinct input vectors, by preventing the additive perturbation from vanishing or exploding. Prior works in mean-field theory Gilboa et al. (2019); Chen et al. (2018a) provide an extensive analysis of a similar objective on the performance of a wide range of recurrent networks.

The Froebenius norm of the temporal Jacobian, defined below, is thus key to both forward and backpropagation. Both processes are significantly expedited when the norm is close to 1.

$$\chi = \frac{1}{N(S-1)}\sum_{t=1}^{S-1}\mathbb{E}(\|\mathbf{J_t}\vec{\mathbf{1}}\|_2^2) = \frac{1}{N(S-1)}\sum_{t=1}^{S-1}\mathbb{E}\left(\sum_{i=1}^{N}\left|\sigma_i^{(t)}\right|^2\right)$$

(3)

## 2.3 PRUNING CRITERIA

Under typical recurrent model initializations, where $\boldsymbol{\theta} \sim \mathcal{N}(\mu_\theta, s_\theta^2\mathbf{I})$ or a similar distribution, with $\mu_\theta \sim 0, s_\theta^2 \ll 1$, Gilboa et al. (2019) has empirically observed that $\chi$ is $< 1$, and that the singular values concentrate towards 0 (see Figure 2 for further evidence). Therefore, we hypothesize that the fastest converging and best performing sparse models are those which simply maximize $\chi$.

We would like to determine the effect of removing one parameter on the Jacobian during the training trajectory. However, as we restrict ourselves only to information available at initialization, we approximate the effect of each parameter on the Jacobian by a first-order Taylor expansion. This is analogous to the derivations given in Lee et al. (2018); Wang et al. (2020):

$$d_n \propto |[\Delta\chi]_n| = \frac{1}{S-1}\sum_{t=1}^{S-1}\left|\frac{\partial}{\partial\theta_n}\|\mathbf{J_t}\mathbf{1}\|_2^2\right|$$

(4)

We call $d_n$ the *sensitivity score* of parameter $\theta_n$.

This criterion will not be well-normed across different types of parameters. This is due to numerous factors, including differing activation functions used for each gate, and differing distributions between the input and recurrent state. Consequently, the variance of our objective is not uniform between groups of parameters (see Section 3.3 for empirical confirmation). We compensate for this by dividing our criterion by the expected magnitude of the gradient for each parameter. The normalized sensitivity score becomes:

$$d_n = [\Delta\tilde{\chi}]_n = \approx \frac{[\Delta\chi]_n}{|\gamma_n|}, \gamma_n = \mathbb{E}_{\tilde{\mathcal{D}}}\left[\sum_{t=1}^{S}\sum_{i=1}^{O}\frac{\partial\tilde{h}_i^{(t)}}{\partial\theta_n}\right]$$

(5)

.

where $\tilde{\mathcal{D}}$ is either the data distribution or an approximate distribution (since we are only trying to estimate the approximate variance of the gradient distribution), and the sequence $\{\tilde{\mathbf{h}}^{(t)}\}$ is computed

on inputs from that distribution. This normalization scheme is similar in motivation to the normalization proposed in Pascanu et al. (2013), and allows us to consider all recurrent models with only one additional computation.

For our pruning objective, we simply take the $K$ weights with largest sensitivity scores, as those represent the parameters which most affect the Jacobian objective near the initialization. Formally, we find the $K$-th largest sensitivity, $\tilde{d}_K$, and set $c_n = I_A(d_n \geq \tilde{d}_K)$. Empirically, we find that the sensitivity score remains an effective metric even if the weights are not restricted to a neighborhood where the Taylor expansion is valid (see Figure 2 for details).

This objective is simple to compute, requiring only two backward passes using auto-differentiation. Furthermore, as we only depend on the Jacobian-vector product, it has a memory cost linear in the parameters.

---

**Algorithm 1** Pruning Recurrent Networks

---

**Require:** Parameters $\theta$, Dataset $\mathcal{D}$, Approximate Dataset $\tilde{\mathcal{D}}$, Sparsity Level $K$, Sequence Length $S$, Number to Sample $P$, Sequence Horizon $U$

1: **for all** $p = 1 \ldots P$ **do**
2:     Sample sequence $(\tilde{\mathbf{X}}, \tilde{\mathbf{Y}}) \sim \tilde{\mathcal{D}}$, $(\mathbf{X}, \mathbf{Y}) \sim \mathcal{D}$
3:     **for all** $t = 1 \ldots S$ **do**
4:         Compute $\{\tilde{\mathbf{h}}^{(t)}\}$ with $(\tilde{\mathbf{X}}, \tilde{\mathbf{Y}})$, $\{\mathbf{h}^{(t)}\}$ with $(\mathbf{X}, \mathbf{Y})$
5:     **end for**
6: **end for**
7: Compute $\boldsymbol{\gamma}$ using $\{\tilde{\mathbf{h}}^{(t)}\}$ and Equation 5
8: **for all** $u = 1 \ldots U$ **do**
9:     Compute $\chi^{(u)} \leftarrow \|\mathbf{J}_{S-u}\mathbf{1}\|_2^2 = \mathbb{E}\left[\sum_{i,j}\left|\frac{\partial h_i^{(S-u)}}{\partial h_j^{(S-u-1)}}\right|^2\right]$
10:     Compute $\Delta\boldsymbol{\chi}^{(u)} \leftarrow \left|\nabla_\theta \boldsymbol{\chi}^{(u)}\right|$
11: **end for**
12: Compute $\mathbf{d} \leftarrow \frac{\sum_t \left[\Delta\boldsymbol{\chi}^{(t)}\right]}{|\boldsymbol{\gamma}|}$
13: $\tilde{d} \leftarrow SortDescending(\mathbf{d})$
14: $c_n \leftarrow \mathbb{1}[d_n \geq \tilde{d}_K], \forall n$
15: **return c**

---

### 2.4 COMPARISON TO EXTANT METHODS

There are two recently proposed criteria for pruning at initialization: GraSP (Wang et al., 2020), and SNIP (Lee et al., 2018). They are given by:

$$\text{GraSP}(\theta) = \boldsymbol{\theta}^T\mathbf{Hg} \tag{6}$$

$$\text{SNIP}(\theta) = \left|\boldsymbol{\theta}^T\mathbf{g}\right| \tag{7}$$

where $[\mathbf{H}]_{ij} = \mathbb{E}[\frac{\partial^2 \mathcal{L}}{\partial\theta_i\partial\theta_j}]$, $\mathbf{g} = \mathbb{E}[\nabla_\theta\mathcal{L}]$ are the expected gradient and Hessian respectively.

Both methods rely on the gradient of the loss with respect to the weights, with SNIP being more dependent on this gradient than GraSP. Thus, the main term of interest is $\mathbf{g}$, which can be decomposed as:

$$\mathbf{g_t} = \tilde{\mathbf{G}}_t \nabla_{\mathbf{h}^{(t)}} \tilde{L}_t \tag{8}$$

With $\tilde{\mathbf{G}}_t$, the Jacobian of $\mathbf{h}^{(t)}$ with respect to $\boldsymbol{\theta}$, as defined in Equation 1.

A consequence of the smaller singular values of $\mathbf{J}$ is that the successive terms of $\tilde{\mathbf{G}}_t$ tend to vanish over time. Thus, loss-based gradient objectives tend to be biased toward explicit dependency between $\mathbf{h}^{(t)}$ on $\boldsymbol{\theta}$, thus neglecting long-term dependence between $\mathbf{h}^{(t)}$ and $\mathbf{h}^{(t-1)}$.

In certain cases, (ex. when the hidden state is small relative to the input) SNIP and GraSP prune many recurrent connections while leaving the input connections largely untouched (see section 3). In contrast, our algorithm considers the $\mathbf{J}$ matrix explicitly, which mitigates the problem of pruning too many recurrent connections.

| Architecture | # of Parameters | Random | Ours | Dense | Δ |
|---|---|---|---|---|---|
| Basic RNN Cell | 171k ↦ 8.5k | 9.51±3.98 | **7.57**±0.20 | 7.08±2.08 | +4.39 |
| Standard LSTM | 684k ↦ 34.2k | 2.17±0.18 | **1.66**±0.16 | 0.80±0.18 | +0.86 |
| Peephole LSTM | 1.32M ↦ 66.2k | 1.80±0.18 | **1.24**±0.08 | 0.74±0.10 | +0.50 |
| GRU | 513k ↦ 25.7k | 1.50±0.08 | **1.46**±0.05 | 0.77±0.14 | +0.69 |

Table 1: Validation Error % of Various 400 Unit RNN Architectures after 50 Epochs of Training on Seq. MNIST; our method works well across all common recurrent architectures. Sparsity of 95 % was used on all experiments.

## 3 EVALUATION

For the following experiments, we compute the $\ell_2$ norm of $\mathbf{J}\vec{1}$ using a single minibatch of 64 data samples, and using only the last 4 steps of the sequence.

### 3.1 SEQUENTIAL MNIST BENCHMARK

We first test our method on the sequential MNIST benchmark (Lee et al., 2018), a relatively small dataset which contains long term dependencies. We begin by verifying that our algorithm is robust across several common recurrent architectures. The results in Table 1 confirm that our method is not dependent on any specific recurrent neural architecture choice.

Our principal results for the Sequential MNIST benchmark are presented in Table 2. Again, we see that our network's performance improves with network size, with the largest gap between our method and others coming when the network grows to 1600 units. We observe that SNIP and GraSP are surprisingly effective at small scales with good initialization, but fail when scaled to larger network sizes. Of the baselines, only random pruning is competitive when scaled, a fact we found quite interesting. For reference, we also provide results on standard L2 pruning (Reed, 1993) (for which the schedule can be found in the appendix) and random pruning. The reader should be careful to note that L2 pruning requires an order of magnitude more resources than other methods due to it's prune – retrain cycle; it is only considered here as a lower bound for network compression. Furthermore, while GraSP requires computing the Hessian gradient across the entire dataset, this is computationally infeasible in our case and we instead compute it with a single minibatch, for fairness.

| Pruning Scheme | 100 Units | 400 Units | 1600 Units |
|---|---|---|---|
| Unnorm. SNiP | 88.9±0.1 | 88.8±0.1 | 89.0±0.1 |
| Norm. SNiP | 4.09±1.06 | 1.52±0.11 | 1.10±0.11 |
| Unnorm. GraSP | 88.6±0.1 | 88.7±0.1 | 88.6±0.1 |
| Norm. GraSP | 4.28±0.57 | 1.62±0.24 | 1.22±0.14 |
| Random | **2.78**±0.25 | 1.50±0.08 | 1.15±0.12 |
| Ours | 3.09±0.31 | **1.46**±0.05 | **1.01**±0.05 |
| L2 | 1.03±0.05 | 0.71±0.03 | 0.57±0.02 |

Table 2: Benchmarking of Various Pruning Algorithms on 95% Sparse GRUs on seq. MNIST. SNIP, GraSP and Random pruning are competitive for smaller models, but the results tend to diminish as the network size increases. Our method obtains strong results even as the network size is large. Further experimental details can be found in the appendix.

In the preceding section, we postulated that normalization across the objective was necessary for strong performance (see Equation 5). This intuition is confirmed in Table 2, where we present both the normalized results (with Glorot Glorot & Bengio (2010) and $\gamma$ normalization) and the unnormalized results (without both). Indeed, we see that this normalization is crucial for recurrent architectures, with unnormalized architectures having all of the retained network weights concentrated in a single gate. This proved to be prohibitive to training.

Finally, in Table 3, we examine the performance of our algorithm at various sparsity levels. Our algorithm continues to outperform random pruning, even at high sparsity levels.

| Sparsity Level (%) | # of Parameters | Random | Ours | Dense | Δ |
|---|---|---|---|---|---|
| 90 | 68.4k | 1.12±0.16 | **1.05**±0.08 | 0.63±0.02 | +0.42 |
| 95 | 34.2k | 1.50±0.08 | **1.46**±0.05 | 0.77±0.10 | +0.69 |
| 98 | 13.7k | 1.82±0.22 | **1.77**±0.07 | 0.67±0.13 | +1.10 |

Table 3: Sparsity Level vs Validation Error % on 400 Unit GRUs, for seq. MNIST. Our method consistently beats random pruning.

## 3.2 Linguistic Sequence Prediction

We assess our models on 3 sequence prediction benchmarks: 1) WikiText-2 (wiki2). 2) WikiText-103 (wiki103), an expanded version of (1) with 10 times more tokens. 3) A truncated version of the One Billion Words (1b) benchmark (Chelba et al., 2013), where only the top 100,000 vocabulary tokens are used. The full experiment parameters are given in the appendix. We report the training and validation perplexities on a random 1% sample of the training set in Table 4.

| Dataset (%) | Random | Ours | Dense | Δ |
|---|---|---|---|---|
| wiki2 | 22.66 | **20.54** | 10.479 | +10.61 |
| wiki103 | 49.65 | **46.65** | 35.87 | +10.78 |
| Trunc. 1b | 59.17 | **53.26** | 38.98 | +14.28 |
| # of Parameters | 960k | 960k | 19.2M | - |

Table 4: Training Perplexities of Training Sparse Models on Large Language Benchmarks. Our method successfully reduces the perplexity score across all benchmarks, often significantly, however there is still a large gap to the dense performance. Parameters are reported only for the recurrent layer as other layers were not pruned during training.

From the results, it is clear that our algorithm succeeds in decreasing perplexity across all language tasks. Despite their varying difficulties, our algorithm speeds up initial convergence on all tasks and maintains an advantage throughout training.

Finally, we perform an ablation experiment on the Penn Treebank Dataset (PTB) with an 800 unit GRU at different sparsity levels. The results are reported in Table 5.

| Sparsity | 0% | 20% | 40% | 60% | 70% | 80% | 90% | 95% | 98% |
|---|---|---|---|---|---|---|---|---|---|
| Perplexity | 156.16 | 160.32 | 165.13 | 173.51 | 178.55 | 184.85 | 194.79 | 208.14 | 228.22 |
| Parameters | 2.88M | 2.30M | 1.72M | 1.15M | 864K | 576K | 288K | 144K | 57.6K |

Table 5: Validation Perplexities of Pruned 800-unit GRU Models on Penn Treebank. For a simple comparison we do not finetune these models, or apply any regularization tricks besides early stopping.

The loss from sparsity increases dramatically as the percentage of parameters remaining approaches zero. This trend is similar to that reported in Gale et al. (2019) and other prior works. For reference, a dense 200 unit GRU (360k parameters) achieves 196.31 perplexity while a 100 unit GRU (150k parameters) achieves 202.97 perplexity.

## 3.3 Qualitative Analysis

The success of our algorithm can be partially attributed to effective distributions across hidden units. Whereas many of the other algorithms are overly concentrated in certain gates and biased towards the input weights, our algorithm effectively distributes sparse connections across the entire weight matrix. We discuss the distribution of remaining connections on a 400 unit GRUs in Figure 1. We also give a set of sample connections under each algorithm in Figure 3.

Finally, we perform an empirical study of the evolution of the Jacobian spectrum to verify our hypothesis on recurrency preservation. We show a 400-unit GRU trained on sequential MNIST, with a dense network, our pruning scheme, and random pruning respectively. It can be observed from Figure 2 after 50000 training steps that our Jacobian has both higher mean and much fewer near-zero singular values, which helps to explain our performance and justify the intuition behind our algorithm. The spectra at initialization also further confirms that the initial singular values of **J** are small.

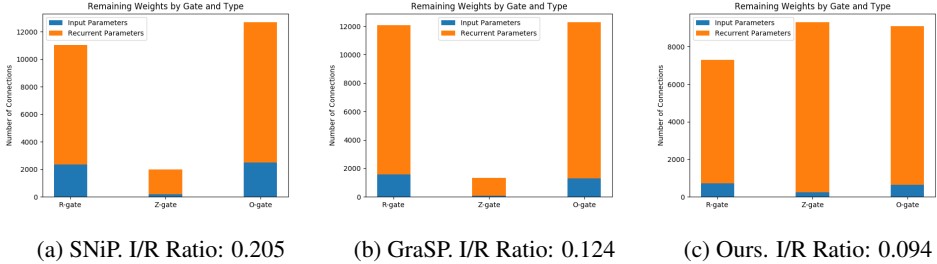

(a) SNiP. I/R Ratio: 0.205     (b) GraSP. I/R Ratio: 0.124     (c) Ours. I/R Ratio: 0.094

Figure 1: Plot of Remaining Connections by Gate and Type. SNiP and GraSP consistently prune recurrent connections at a much higher ratio than input connections. The ratio of remaining input to recurrent (I/R) connections is given for each method; the dense ratio is 0.07 for comparison. SNiP and GraSP also exhibit severe imbalance between gates, while our imbalance is far milder.

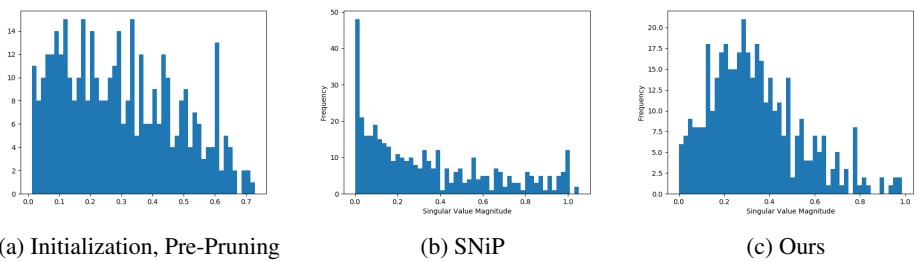

(a) Initialization, Pre-Pruning     (b) SNiP     (c) Ours

Figure 2: Singular Value Magnitude Histograms after 50 epochs of Training, for 400 Unit GRU on seq. MNIST. Compared to SNiP, our method prevents spectral concentration at 0, with a mean singular value magnitude of 0.31 to SNiP's 0.18 This helps to explain our relative performance gain.

## 4 OTHER RELATED WORK

**Methods for Pruning Recurrent Networks:** Our method is the latest in a series of attempts to generate sparse RNNs. Perhaps the most well-known algorithm for sparse network pruning is Narang et al. (2017a). It is a modification to magnitude based pruning wherein the pruning threshold evolves according to several hyperparameters that have to be tuned by the user. Kliegl et al. (2017) uses iterative trace norm regularization to prune RNNs used for speech recognition. This effectively reduces the sum of the singular values of the weight matrices. But we found in our experiments that these values were often degenerate near 0. Furthermore, this technique is iterative. Narang et al. (2017b) uses iterative ground lasso regularization to induce block sparsity in recurrent neural networks. Wen et al. (2017) alters the structure of LSTMs to decrease their memory requirements. Their intrinsic sparse structures make structural assumptions about the sparsity distribution across the network. Dai et al. (2018) uses magnitude based pruning coupled with a special RNN structure

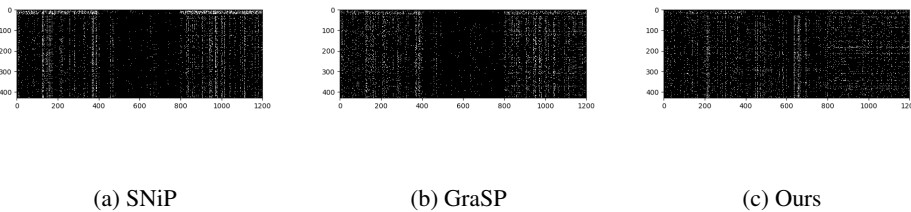

(a) SNiP     (b) GraSP     (c) Ours

Figure 3: Map of remaining connections, with the x-axis indicating the output size (flattened across gates) and the y-axis indicated the input size.Our method is significantly more spread out across neurons and gates than the others.

to make RNNs more efficient. The pruning algorithm itself is magnitude based. See et al. (2016) uses iterative pruning and retraining to prune a recurrent model for neural translation. The underlying technique is simple iterative pruning, and the final pruning percentage is only 80%. While fine for their application, we are interested in novel pruning techniques and higher levels of sparsity.

In summation, all the the methods discussed above utilized some variant of L1 or L2 pruning to actually sparsify the network. The novel advances are all related to pruning schedules, modifications to recurrent architectures, or small transformations of the L1 or L2 objective.

**Other Pruning Techniques:** Many extant pruning techniques are applicable to recurrent network architectures, even if these methods were not designed from the ground up to work in the recurrent case. Lee et al. (2018) and Wang et al. (2020) both provide a pruning objective that can be used to prune networks before training begins. They are considered extensively in this work. In Frankle & Carbin (2019), it is shown that at initialization networks contain a small sparse set of connections that can achieve similar results to fully dense networks. However, no known method yet exists to recover these sparse networks to the full extent demonstrated in that work. Han et al. (2015) showed impressive results with magnitude based pruning. Follow up work made further use of magnitude-based pruning techniques (Carreira-Perpinán & Idelbayev, 2018; Guo et al., 2016); however, these techniques are primarily iterative.

Mean Replacement Pruning (Evci et al., 2018) uses the absolute-value of the Taylor expansion of the loss to as a criterion for which units in a network should be pruned. This method can not be used with BatchNorm and achieves results comparable to magnitude based pruning. Bayesian methods have recently seen some success in pruning neural networks. (Ullrich et al., 2017), which is itself an extension of Nowlan & Hinton (1992), is the standard citation here. In essence, this method works by re-training a network while also fitting the weights to a GMM prior via a KL penalty. Molchanov et al. (2017) is another Bayesian pruning technique that learns a dropout rate via variational inference that can subsequently be used to prune the network. Finally, there exists several classical pruning techniques. Ishikawa (1996); Chauvin (1989) enforced sparsity penalties during the training process. LeCun et al. (1990); Hassibi et al. (1993) perform Hessian-based pruning, using the Hessian to get a sensitivity metric for the network's weights.

While many of the above methods are effective in general, they do not explicitly consider the specifics of RNNs and sequential prediction.

**Other Related Work** Several interesting papers have recently taken a critical look at the problem of network pruning (Liu et al., 2018; Crowley et al., 2018). The problem of network compression is closely related to network pruning. It would be impossible to cite all of the relevant papers here, and no good literature survey exists. Some worthwhile references are Gupta et al. (2015); Gong et al. (2014); Courbariaux et al. (2016); Chen et al. (2018b); Howard et al. (2017). Both problems often share a common goal of reducing the size of a network. Some notable papers explicitly consider the problem of recurrent network compression (Ye et al., 2018; Lobacheva et al., 2017; Wang et al., 2018).

In the context of the above work, our method is not iterative and can be fully completed before training even begins. The tradeoffs in accuracy can be remedied by scaling up the network, since there is no longer a need to store fully dense weights during training. Furthermore, our objective is specifically adapted to the sequential prediction context in which RNNs are deployed. We are the first pruning algorithm to consider the temporal Jacobian spectrum as a key to generating faster converging and better performance sparse RNNs. Our method not only performs better in practice compared to other zero-shot methods, but also yields key insight into the factors behind RNN performance. This may aid the development of new architectures and training schemes for sequential prediction.

## 5    CLOSING REMARKS

In this work, we presented an effective and cheap single-shot pruning algorithm adapted toward recurrent models. Throughout the work, we continually found the importance of the Jacobian spectrum surprising and interesting. Future work could further examine the relationship between network width, the Jacobian spectrum, and generalization.

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

# 6 APPENDIX A - EXPERIMENT HYPERPARAMETERS

Unless otherwise specified, our model consists of a single-layered RNN, followed by an appropriately sized softmax layer with sigmoidal activation. The softmax layer is initialized with standard Xavier. We use a minibatch size of 64 samples during training, and optimize using the AdaM optimizer (Kingma & Ba, 2014) with a learning rate of 1e-3. We use an initial hidden state of zeros for all experiments.

For all networks, we only prune the recurrent layer while leaving prior and subsequent layers untouched, since we are primarily interested in performance of recurrent layers. We trained all networks with a single Nvidia P100 GPU.

## 6.1 SEQUENTIAL MNIST

For seq. MNIST, we follow the same process as SNiP, feeding in row-by-row. We used $\mathcal{N}(0, 0.1)$ for our own method, and Glorot initialization for SNiP and GraSP. $\gamma$ is computed from data sampled from a $\mathcal{N}(0, 0.1)$ distribution. We use only the activations from the last time step. For L2, the density was annealed according to the schedule $\{0.8, 0.6, 0.4, 0.2, 0.1, 0.05, 0.02, 0.01\}$ every 10k training steps.

## 6.2 LANGUAGE BENCHMARKS

We use 2000-unit LSTMs for all language benchmarks. To reduce the variance of our comparison, we freeze the embedding layer before training. We use sampled sequential cross-entropy loss with 1000 tokens for wiki103 and 1b, and standard cross-entropy for wiki2. We use He initialization for all papers.

Wiki2 was trained for 20k training steps (13 epochs), while wiki103 was trained for 12k training steps, and 1b was trained for 30k training steps.

# 7 APPENDIX B - ADDITIONAL STUDIES

## 7.1 INITIALIZATIONS

We benchmark the performance of our algorithm against random pruning using 3 additional initializations, seen in Table 6. With high variance, the first-order expansion we use to estimate our objective fails to hold, so we do significantly worse than the random benchmark.

| Initialization Scheme | Ours | Random |
|---|---|---|
| Glorot | **1.219** | 1.36 |
| $\mathcal{N}(0, 1)$ | 3.30 | **1.38** |
| $uniform(0, 0.1)$ | 1.73 | **1.32** |

Table 6: Benchmarking of validation Error % on Different Initializations, for Sequential MNIST Task with 400 Unit GRU. Our algorithm successfully beats random on well-conditioned normal distributions, but fails on high variance and the uniform distribution.

## 7.2 RUNTIME

We benchmark the runtimes of SNiP, GraSP and our own algorithm, using only a single batch and time iteration for fairness, seen in Table 7.

# 8 APPENDIX C - TRAINING CURVES

We present a sample training curve of a 400 unit GRU for sequential MNIST below. As can be seen, random pruning is only competitive algorithm in this instance.

| Pruning Scheme | Runtime (seconds) |
|---|---|
| SNiP | **4.718** |
| GraSP | 16.406 |
| Ours | 4.876 |

Table 7: Benchmarking of Pruning Algorithm Runtimes; our method is faster than GraSP as the Hessian is larger than the Jacobian, but slower than SNiP for a single time instance. It should be noted that our algorithm works best when iterated across several time steps, while GraSP requires iteration across the entire training set, and SNiP requires only a single computation.

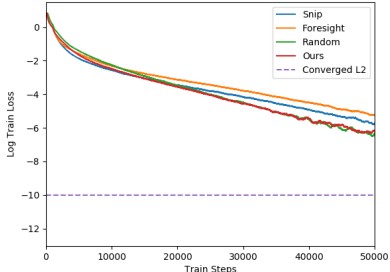

Figure 4: Plot of Log Train Loss for a 400 Unit GRU, trained on Sequential MNIST. GraSP is the worst performing, followed by SNiP and then Random, which is on par with our method. L2 is shown as a lower bound. It is surprising that random is competitive, but it is free from the gate imbalance exhibited by SNiP and GraSP.

Subsequently we present a sample training curve in Figure 5 for the 1b words experiment, detailed in Table 4. Our algorithm provides significant benefit over random pruning, but still lags behind the dense model.

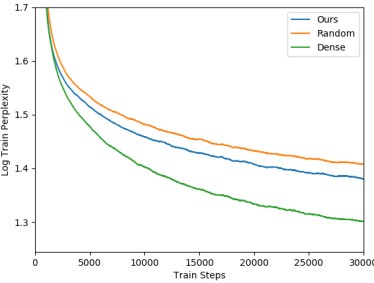

Figure 5: Plot of Log Train Perplexity on the 1b dataset, with 2k LSTM network. Our model clearly outperforms random pruning by a significant margin, however more work is needed before we achieve near-dense performance.

