# OpenReview forum: "One-Shot Pruning of Recurrent Neural Networks by Jacobian Spectrum Evaluation"
_ICLR.cc/2020/Conference — Accept (Poster)_

### Official Review · AnonReviewer2 · 2019-10-08
**Official Blind Review #2**

**Rating:** 6

**Review:**

OVERALL:

I should first say that this is reasonably far outside my wheelhouse.
I have never worked on RNNs or pruning.
I also have no familiarity with the data sets used.

All these things being said, I can follow the derivations, and the idea
seems reasonable and well-motivated, and pruning is interesting
for both scientific and practical reasons, and this technique seems to help a substantial amount,
so I'm inclined to vote for acceptance, with the understanding that perhaps better informed reviewers
 will in the future point out something I have missed.

DETAILED NOTES:

> overparameterized networks require more storage capacity and are computationally more expensive than their pruned counterparts
I'm with you on the storage capacity, but do any of these pruned networks actually run faster than their non-pruned counterparts?
I thought you had to work really hard to prune to some kind of block-sparse representation to realize any speed gains.
This question is not rhetorical - I know very little about this topic.

> Work for the present volume began by asking the question...
I like this paragraph for motivation, but perhaps 'volume' is slightly overwrought?

> For our pruning objective, we simply take the
K weights with largest sensitivity scores, as those represent the parameters which most affect the
Jacobian objective near the initialization.
Is there some notion of redundancy, where certain sets of parameters affect the jacobian in the same way,
so that all but 1 element of the set could be pruned, or something?

In line 13 of algorithm 1, why do we need to sort if in the next step we just mask out
everything with sensitivity less than D tilde k?

Maybe this is a dumb question, but if you're pruning at initialization, why not
just initialize a smaller network in such a way that you wouldn't choose to prune any of its parameters?
Am I misunderstanding what you're doing?


> In the prequel, we postulated t
The prequel?

Fig 1 is interesting, but it raises the question of whether you could recover a simpler
algorithm by just modifying random pruning so that it evenly distributes 'prunes' across
gate and Type.

In fig 2, why are all the singular values less than 1?
It's not obvious to me why that should be true, unless you enforce it w/ the initialization.


**Experience Assessment:**

I do not know much about this area.

**Review Assessment: Checking Correctness Of Derivations And Theory:**

I assessed the sensibility of the derivations and theory.

**Review Assessment: Checking Correctness Of Experiments:**

I assessed the sensibility of the experiments.

**Review Assessment: Thoroughness In Paper Reading:**

I read the paper at least twice and used my best judgement in assessing the paper.

---

> ### Author Response · Authors · 2019-11-08
> **Response to Reviewer 2**
>
> We thank the reviewer for their insights and comments.
>
> We are happy to know that you found the paper well-motivated and were able to follow the derivations. We put substantial effort into making the derivations easy to follow. It is great feedback to learn that these efforts were worthwhile.
>
> With respect to each of your comments:
> Runtime of Sparse Networks: At high sparsity (above 90%), pruned networks typically run faster than their dense counterparts even without adaptions such as block-sparsity. Tensorflow and other standard libraries have operations specifically dedicated to fast sparse multiplication and autodifferentiation. Thank you for calling attention to this.
>
>
> Redundancy in Parameters: Although there may be groupings of similar parameters with respect to Jacobian contribution, capturing second-order correlations is nontrivial and adds significantly to the runtime of the algorithm. We may explore such ideas in a sequel of our work.
>
> Initializing a smaller sparse network: Even with random pruning, we find that large sparse networks are more parameter efficient than small dense networks. For example, on PTB, a 95% sparse network achieves 47.2 train perplexity, compared to an equivalently parameterized dense network performance of 48.4. Consequently, there is a need to efficiently sparsify RNNs, even if we observe a drop in performance before and after sparsification.
>
> Comments on Figure 1: Random pruning is already evenly distributed across gates and type (for a 400 unit GRU with 500k parameters, the unevenness across gates and types is statistically negligible). This is precisely why random pruning obtains such strong performance compared to our algorithm. However, homogeneity across gate/type is not the sole contributor to our algorithm's success, as shown in our paper's analysis.
>
> Comments on Figure 2: For standard, variance-preserving initializations (He et. al.), the singular values tend to be less than 1. For more irregular, high variance distributions, this assumption does not hold. Our algorithm performs poorly on such distributions, as shown in the appendix.
>
> Stylistic feedback has been incorporated in a revision of the document. We apologize for any distraction this caused. We were having fun with the writing but were perhaps too overwrought at times.

---

### Official Review · AnonReviewer1 · 2019-10-09
**Official Blind Review #1**

**Rating:** 6

**Review:**


Notes:

  -RNN network pruning has proven to be challenging using the techniques often used with other network types.

  -One issue is that the performance of an LSTM/GRU can hinge on a few activated gates and can lead to more concentration of influence than would be seen in a feedforward network without parameter sharing between layers.

  -New objective uses 64 points to prune the network (I assume this is just the size of the minibatch).

  -Result is a 95% sparse GRU cell.

  -New idea is based on keeping weights that propagate information through many time steps.

  -Encourages "singular values of the temporal jacobian with respect to network weights to be non-degenerate" (I suppose this means that the gradient flowing through time will contain multiple directions of variation)

  -Introduction does a good job of introducing the key ideas.

  -J_t is a temporal jacobian of size N x N (N is number units) at time step t.

  -Chi is the spectral norm of this temporal jacobian.  I'm a bit confused by this, because my understanding is that the spectral norm is the largest singular value, but equation 3 looks like a sum over singular values, making it more like a frobenius norm?

  -This jacobian isn't tractable, so paper approximates it using a first-order taylor expansion.  So basically the pruning just amounts to taking parameters with the largest gradient?

  -Section 2.4 is confusing and seems to come out of nowhere.  Is this suggesting that the technique isn't just pruning but adding a new normalization scheme?  On second reading, this is a normalization scheme effecting which parameters to prune.  The motivation for why the gradients are normalized like this is still confusing.  If you're willing to make a linear assumption, it seems like it's enough to consider the gradient on the parameter multiplied by the magnitude of the parameter to see the overall effect of removing it?

  -The results look good, but sequential mnist is a bit of a toy task.  I'd also like to see a more fine-grained analysis showing the tradeoff between the number of units removed and the performance.

  -The paper claims that L2 pruning requires more data, but it's unclear if this really matters since the whole dataset was used to train both methods initially.

  -On Table 2 the results of the technique don't seem that much better than "Random".

  Review: This paper presented a fast pruning algorithm for RNNs, which uses the norm of the gradient as a guide to pruning.  I'm borderline on this paper.  The idea of using the gradient is good, but the explanation of some aspects like the normalization is confusing and felt random.  Additionally the results, while better than some other pruning techniques on RNNs, don't seem to be that much better than random.

**Experience Assessment:**

I do not know much about this area.

**Review Assessment: Checking Correctness Of Derivations And Theory:**

I assessed the sensibility of the derivations and theory.

**Review Assessment: Checking Correctness Of Experiments:**

I assessed the sensibility of the experiments.

**Review Assessment: Thoroughness In Paper Reading:**

I read the paper at least twice and used my best judgement in assessing the paper.

---

> ### Author Response · Authors · 2019-11-08
> **Response to Reviewer 1**
>
> We thank the reviewer for their careful analysis of our paper. The review's insightful comments helped us to substantially improve the clarity of our work. We would also like to thank you for speaking about the strength of our experimental results (with some interesting caveats, of course, about random pruning).
>
> The reviewer has enumerated three primary hesitations with our paper. We would like to respond to those as follows:
>     1. "the explanation of some aspects like the normalization is confusing and felt random"
>
>     Architecture choices such as the activation function applied to each gate will affect the resulting variance of each parameter's score, irrespective of actual contribution to the Jacobian spectrum. Nonetheless, we agree that the importance of this section was overstated in our original draft, and that the motivation could be further elaborated. The updated paper no longer emphasizes this section (see the end of section 2.3).
>
>     2. "Additionally the results, while better than some other pruning techniques on RNNs, don't seem to be that much better than random."
>     The performance of random pruning has been surprising, and is the focus of a good deal of analysis in our paper. When discussing random pruning of recurrent networks, it is crucial to keep in mind that most algorithms that are not adapted to RNNs fail to improve upon random pruning. Indeed we are the first algorithm to systematically improve on random pruning for sequential models, especially at large parameter scales. We hope to see future papers revisit this topic, and explore the performance of random pruning more deeply.
>
>     3. "I'd also like to see a more fine-grained analysis showing the tradeoff between the number of units removed and the performance."
>     We have included an ablation experiment on Penn Treebank showing the tradeoff between parameters removed, and performance of the resulting network. This supplements the analysis performed on sequential MNIST. See the revised paper for more comments (end of section 3.2).
>
> We would also like to respond to several of the smaller concerns that were raised.
>
>  - Pruning amounts to taking parameters with the largest gradient:
>  Our method estimates the parameter impact on the Jacobian singular values through first order expansion. While the Jacobian itself is tractable, optimizing it with binary search on sparse masks is not. Hence, we take the gradient of the Jacobian as our approximation. Our method is second-order with respect to the data and parameters.
>
>  - L2 Pruning: We intended to state that L2 pruning required significantly more resources during training than our method. The ambiguity in our paper has been clarified (second paragraph of 3.1).
>
>  - Stylistic Errors: These have been corrected in a revision of the paper. We thank the reviewer for taking the time to point these out.
>
> Please let us know if we have addressed the review's concerns enough to merit a change in score. If you are still borderline about our paper clearing the bar for acceptance, please let us know if there's any other analysis that could help.

---

> > ### Author Response · Authors · 2019-11-13
> > **Additional Feedback**
> >
> > Dear Reviewer 1, we wanted to see if you had a chance to review the changes we've made to the paper. We think we addressed many of your concerns, and would be interested in hearing any other useful feedback you might have.

---

> > ### Comment · AnonReviewer1 · 2019-11-14
> > **Response to rebuttal**
> >
> >
> > 1.  I reread the new section 3.2.  I think it's much better now, as it is very clear that the pruning criteria is a ranking over sensitivity scores.  In the original draft I found that part quite confusing.
> >
> > 2.  The point about other pruning methods failing for RNNs makes sense to me.  I'm still wary of a paper which has results which are really not that much better than random.  Part of .  Although for sequential mnist for 1600 units, the improvement just barely exceeds 1 stdv.  Thus it's somewhat unlikely to just be random variation between experiments.
> >
> > Overall I'm still quite borderline, because I feel like another round of refinement could improve the paper.  At the same time, I think the 3/10 score is too low, and I do think that rejecting this paper could hold back progress in this area.  So I'll raise my score to 6/10.

---

> > > ### Author Response · Authors · 2019-11-14
> > > **Re:Response to rebuttal**
> > >
> > > Thank you for your thoughtful comments and taking the time to review our rebuttal. We really appreciate it, and we will continue to refine our results in the remaining time.

---

### Official Review · AnonReviewer3 · 2019-10-23
**Official Blind Review #3**

**Rating:** 6

**Review:**

Although I worked and published on network pruning in the past, I found little overlap between my knowledge and the content of this paper. I feel therefore that I can only provide a very superficial feedback.

The authors generalize recently-developed network pruning techniques that were developed in he context of feed-forward networks, and do not require iterative pruning-retraining cycles, to RNNs. Since RNNs apply the same matrix on their hidden vector multiple times, approximate closed-form expressions for pruning criteria can be derived analytically. The pruning criterion is based on the spectrum of the Jacobian matrix of the (N+1)-th hidden unit with respect to the N-th one. A closed-form algorithm for pruning is presented, and the method is evaluated on several datasets.

While I appreciate the neatness of deriving closed-form analytical expressions for a fast and simple pruning method, I feel that am not in a position to rate the paper. My apologies to the Authors and the Editors.

**Experience Assessment:**

I do not know much about this area.

**Review Assessment: Checking Correctness Of Derivations And Theory:**

I did not assess the derivations or theory.

**Review Assessment: Checking Correctness Of Experiments:**

I did not assess the experiments.

**Review Assessment: Thoroughness In Paper Reading:**

I read the paper at least twice and used my best judgement in assessing the paper.

---

> ### Author Response · Authors · 2019-11-08
> **Response to Reviewer 3**
>
> We thank the reviewer for their generous assessment of our paper. We agree with their analysis, and appreciate their comment on "the neatness of deriving closed-form analytical expressions for a fast and simple pruning method". This, in our opinion, is one of our work's main strengths.
>
> We believe that this reviewer's endorsement is a worthwhile sign, since it means that researchers without extensive knowledge of this particular problem are able to follow our paper's main arguments.

---

### Decision · Program_Chairs · 2019-12-19

**Decision:**

Accept (Poster)

**Comment:**

Based on current unanimous reviews, the paper is accepted.